# SEQUENCE-SOD: SEQUENCE-AWARE SPIKING OBJECT DETECTION FOR EVENT CAMERAS

## ABSTRACT

Due to the asynchronous sensing of changes in illumination by event cameras, they are highly energy-efficient and therefore exhibit great potential especially in mobile, low power scenarios. Moreover, they are able to acquire sparse data with a high temporal resolution in the order of milliseconds and achieve a large dynamic range. This enables the recording of reliable data with minimal motion blur even during rapid movements and in low light scenarios. SNNs are particularly suitable for the processing of event data due to their asynchronous and spike-based functionality while their low energy consumption enables their deployment in automotive embedded applications. However, recent spiking object detectors do not leverage the full temporal information and only consider a single, fixed-size sample of the event data. In this paper, we propose the first sequence-aware SNN, which processes long sequences of the event stream data and predicts bounding boxes with a frequency of 40 Hz. In combination with a SSD network design, we achieve a 26.88 mAP on the Gen1 Automotive Detection Dataset.

## 1 INTRODUCTION

Particularly in highly dynamic scenes, standard frame-based cameras reveal their poor performance in terms of motion blur and nighttime sensing. Their static frame rate further hinders capturing rapid movements of cars and trains, which can cover meters in mere milliseconds. This poses a critical limitation, especially in autonomous driving scenarios where such delays may result in fatal consequences. Therefore, event cameras recently emerged, which neglect information about the absolute intensity and instead asynchronously record logarithmic intensity changes in order to achieve a high temporal resolution in the magnitude of microseconds. They further cover a high dynamic range (about 140 dB compared to 60 dB by standard cameras), which allows the detection of movements even in low light settings. Combined with their low power consumption, this makes event cameras highly suitable for the usage in automotive edge applications.

However, the inherent sparsity and temporal nature of the event data makes it difficult to apply traditional frame-based object detection approaches. A more natural approach to processing events involves the use of Spiking Neural Networks (SNNs) (Gerstner & Kistler, 2002), which communicate using sparse and asynchronous 1-bit spikes. Consequently, SNNs can replace multiplications in a traditional Artifical Neural Network (ANN) with additions or simple memory readouts, resulting in a reduced number of machine operations. This makes SNNs very compelling at a time when energy consumption is a critical issue, particularly in autonomous vehicles.

Real world scenarios demand continuous data stream processing, making it advantageous to utilize information from previous time steps for future object detection. SNNs preserve the system's state by encoding it as the membrane potential within each spiking neuron. If no input current arrives, this potential decays over time, which means that SNNs are able to naturally handle cases in which no events are detected. However, current spiking object detectors, like the one by Cordone et al. (2022), only consider the time steps in a single time interval and then reset the internal state of the SNN after one prediction, which is illustrated in Figure 1a. Instead, we propose **Sequence-SOD**, our sequence-aware approach trained on extended sequences encompassing multiple labels across different time points in event data. This method preserves the spiking neurons' inner states, as depicted inFigure 1b. Theoretically, this enables us to predict bounding boxes at every time step resulting in a remarkable frequency of about 40 Hz.

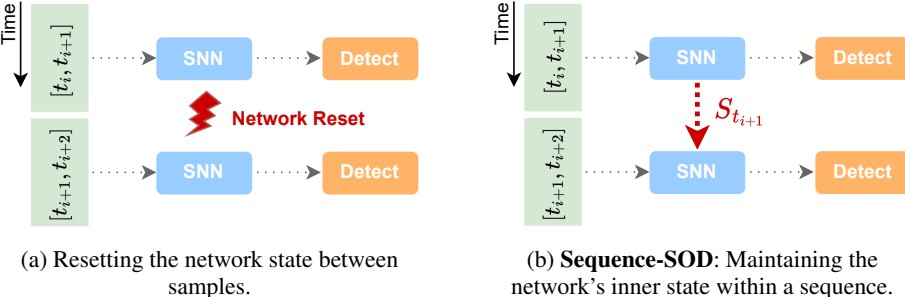

(a) Resetting the network state between samples.

(b) **Sequence-SOD**: Maintaining the network's inner state within a sequence.

Figure 1: Resetting methods of the inner state of SNNs.

The following summarizes the key contributions of this paper:

- We introduce a novel sequence-aware training approach for object detection using SNNs, which considers whole sequences of event data including multiple labels at different points in time, in order to be more performant in real time applications.
- Our approach is able to predict bounding boxes on real world automotive event data with a frequency of $40\,Hz$.
- We evaluate the influence of event data augmentation on the performance of SNNs in object detection, as well as compare the energy consumption of our SNN architecture with the equivalent ANN.

## 2 RELATED WORK

### 2.1 SPIKING NEURAL NETWORKS

The work of Maass & Markram (2004) has shown that in theory SNNs have exactly the same expressive power as ANNs. In addition, they are not only more energy efficient but also show a greater potential in handling event data compared to ANNs (Deng et al., 2019).

When modelling SNNs, the most common design is the Leaky Integrate-and-Fire (LIF) neuron model, whose discretized variant can be described in the following way (Wu et al., 2019):

$$U[t] = \underbrace{\beta U[t-1]}_{\text{decay}} + \underbrace{W X[t]}_{\text{input}} - \underbrace{S_{out}[t-1]\theta}_{\text{reset}}.$$ (1)

The spiking functionality integrates the weighted sum of the incoming spikes $W X[t]$ into the membrane potential $U$ of the spiking neuron. $U$ exponentially decays with a rate $\beta$ to decrease the influence of temporally uncorrelated data. When the membrane potential reaches a certain threshold $\theta$ an output spike is generated and $U$ is reset accordingly.

The disadvantage of this spiking behavior is that it is non-differentiable and therefore the common backpropagation algorithms from ANN training are not directly transferable. So far, three alternative approaches for the challenging training of SNNs have been proposed: Biology-inspired training, conversion from ANNs, and *direct training*.

Biologically plausible methods, such as Spike-timing-dependent plasticity (STDP) (Toyoizumi et al., 2004), adjust the weights of synapses based on the correlation of the spike behavior of pre- and post-synaptic neurons. However, the localization of updates makes it difficult to train deep networks, rendering biological approaches insufficient for complex tasks such as object detection.

Another option is the conversion of a pre-trained ANN into a SNN by weight-normalization, as there is a correlation between the ReLU activation and the firing rate of a spiking neuron (Rueckauer et al., 2017). However, this requires the input to be rate encoded, which means that the number of spikes has to be proportional to the pixel intensity. Consequently, achieving approximately the same accuracy as the pre-trained ANN requires very high latency, defeating the goal of low energy

consumption (Bendig et al., 2023). Furthermore, event data is not suitable as input for ANNs due to its inherent sparsity and asynchronous nature.

The most promising approach is *direct training* using surrogate gradients (Neftci et al., 2019). To overcome the non-differentiability of the binary spikes, the gradient is approximated during the backward pass. This also has the advantage of defining a small gradient even when no spike occurs, ensuring the presence of a training signal in such scenarios.

## 2.2 EVENT-BASED OBJECT DETECTION

The main challenges in event-based object detection lie in the sparsity and asynchronous nature of event data, which pose major difficulties for standard ANNs as well as current clock-based Graphic Processing Units (GPUs).

Common approaches for event-based object detection, especially using conventional ANNs, encompass techniques such as: *Event frame* (an accumulation of events either over a certain period of time or number of events) (Zhao et al., 2022; Shair & Rawashdeh, 2022), *Time-surface* (storing timestamp information of the events for each location) (Mitrokhin et al., 2018) or a custom input layer (weighted accumulation of events over time, based on convolution) (Cannici et al., 2019). However, these approaches are limited to the processing of one event sample at a time due to the incapability of standard ANNs to maintain an inner state over time. Consequently, during the processing of an event stream, previous temporal information will get lost. This limitation leads us to choose utilizing a SNN instead.

The Recurrent Event-camera Detector (RED) (Perot et al., 2020) incorporates ConvLSTM cells and feed-forward convolutional layers in their recurrent architecture, allowing them to retain temporal information within one sequence. Other works Gehrig & Scaramuzza (2023); Li et al. (2022) also integrate recurrent layers and attention mechanisms to process sequences of event data. However, these methods only partially include recurrent neurons, limiting their expressiveness concerning the inner state. Additionally, these works process multiple time steps of one time interval simultaneously, diminishing the potential prediction frequency that event streams allow for. In our paper we opt for a fully spiking architecture, which contains spiking neurons at each layer. This choice does not only lead to an increased capacity for storing temporal information, but also allows for the sequential processing of each time step.

SNNs naturally process asynchronous and sparse data and are capable of storing a network state in the membrane potential of the spiking neurons. They handle small time steps of event data sequentially, allowing a high prediction frequency. The authors of Cordone et al. (2022) present a complete spiking neural network for object detection. Their method, however, only considers individual time intervals as samples, not utilizing the full potential of event sequences during training. In order to fully leverage the temporal information in the event stream, our approach considers extended event sequence samples, encompassing multiple annotations at various time points. This approach not only aligns more closely with real world automotive applications but also allows a rapid prediction frequency of 40 Hz. Additionally, our use of a SNN retains the current network state's information while ensuring low energy consumption for potential embedded applications.

## 3 METHOD

Seeking to fully leverage the temporal resolution of event data, our object detection approach processes entire sequences of events containing multiple labels at different time steps, rather than processing individual event samples one at a time. In our approach, events are initially accumulated into small time intervals. Each event interval is then divided into 5 equally-sized time steps of which are fed to the network sequentially. The output is averaged over 5 time steps for the bounding box and class prediction. Notably, the membrane potentials of the spiking neurons are not reset between samples (time intervals), which allows previous information to be maintained and thus facilitates high frequency predictions after each time step.

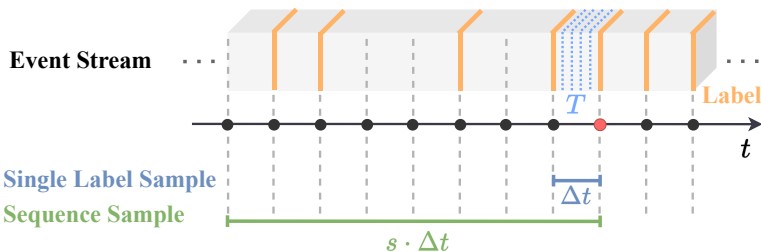

Figure 2: Visualization of different sampling techniques. Our work operates on entire sequences, instead of individual samples.

## 3.1 SEQUENTIAL EVENT DATA PROCESSING

The pixels of an event camera register logarithmic intensity changes in an asynchronous manner. An event occurring at the pixel position $(x_i, y_i)$ at time $t_i$ is commonly defined as a tuple $e_i = (x_i, y_i, t_i, p_i)$. Depending on if the brightness increases or decreases, events have either a positive or negative polarity $p_i \in \{1, -1\}$.

Due to the challenges associated with training deep SNNs on neuromorphic hardware, we simulate the computations in form of time steps on a GPU. Consequently, processing event data asynchronously becomes computationally infeasible, leading us to accumulate events in short time intervals $\Delta t = [t_a, t_b)$ of 125 ms.

Inspired by the event data handling in Gehrig & Scaramuzza (2023), we choose a simple preprocessing method creating event histograms $E$ in the shape $(T, 2, H, W)$ with $T$ representing the number of time steps discretized from $\Delta t$ and $(H, W)$ denoting the height and width of the event sensor. We choose to not further divide the time steps into time bins, since Cordone et al. (2022) showed that time bins do not improve the performance for $T \geq 5$. Moreover, we accumulate events of different polarities separately in two channels. A stream of events $\mathcal{E}$ is thus processed in the following way:

$$E(\tau, p, x, y) = \sum_{e_i \in \mathcal{E}} \delta(\tau - \tau_i)\delta(p - p_i)\delta(x - x_i)\delta(y - y_i)$$

$$\tau_i = \left\lfloor \frac{t_i - t_a}{t_b - t_a} \cdot T \right\rfloor,$$

(2)

with $\delta(\cdot)$ as the Kronecker delta function.

Additionally, we consider whole sequences of the event stream during the training with a sequence length of $s$, which is visualized in Figure 2. One training sequence includes $s$ time intervals, each containing an event histogram with $T = 5$ time steps over $\Delta t = 125ms$. Since bounding box labels in our selected dataset are generated based on the corresponding RGB images, they are intermittently available rather than continuously present, appearing at different points in time with varying frequencies. Consequently, not all time intervals within an event sequence contain bounding boxes. We ensure that at least one label is present in each training sequence. The inner state of an SNN, reflected in the membrane potential of its neurons, retains information from prior event inputs. This stored information, temporally correlated with subsequent inputs, contributes to future predictions. Unlike prior spiking object detection methods that reset the membrane potential after each label occurrence, our SNN maintains its inner state consistently throughout the processing of sequences with multiple labels at varying time points. Previous methods failed to harness the full potential of temporal information within event data. Processing whole sequences compels the network to prioritize essential temporal cues. Individual spiking neurons have a finite capacity and as the input sequence lengthens, they must process an increasing volume of incoming data over time. Consequently, the network is forced to learn a more meaningful inner state representation of the current scenario, discarding redundant information to process new stimuli. Multiple labels at different points in time prompt the network to generate accurate predictions not only at the sequence's end but after every time step within it. This compels the network to consistently maintain an expressive inner state representation, facilitating ongoing predictions in an continuous stream of information.

## 3.2 NETWORK ARCHITECTURE

For consistent comparability, our approach follows the methodology of Cordone et al. (2022) by adopting the SSD object detection framework (Liu et al., 2016) with a Spiking DenseNet (depth of 121 and a growth rate of 24) as its backbone. The dense connectivity pattern of DenseNet (Huang et al., 2017) enhances the propagation of gradients, which is especially important for SNNs since the gradient can only be approximated. Following the example of Cordone et al. (2022), we replace the ReLU activations with Parametric Leaky Integrate-and-Fire (PLIF) (Fang et al., 2021) neurons, which in contrast to conventional LIF neurons include the decay rate $\beta$ as a trainable parameter.

The DenseNet architecture includes batch normalization, which seems to play a crucial role in the training of our SNN. However after the training, it is possible to construct a mathematically equivalent network that absorbs the batch normalization parameters into the convolutional layers (Jacob et al., 2018). Therefore, our network can be seen as a pure SNN during inference, leveraging all the advantages in terms of energy efficiency of the spiking computations.

In contrast to previous recurrent object detectors, which process all time steps simultaneously as an input of shape $(2T, H, W)$, our spiking approach processes an input tensor of shape $(2, H, W)$ at each time step. The final prediction of the bounding boxes and classes is calculated by accumulating the network output over $T$ time steps and dividing it by $T$. Beyond the initial five time steps (equivalent to 125 ms), this can be done with a sliding window approach for the rest of the sequence. Consequently, the network is theoretically able to output bounding box predictions after every 25 ms time step (= 40 Hz), effectively leveraging the high temporal resolution of the event camera. However, the prediction frequency relies on the input data and may improve with smaller time steps, albeit potentially impacting performance negatively.

## 4 EXPERIMENTS AND RESULTS

### 4.1 SETUP

**Dataset**    As we focus on object detection on event data in an automotive setting, we test our method on the Gen1 Automotive Detection Dataset (de Tournemire et al., 2020). The dataset contains 39 h of event camera recordings from diverse driving scenarios with a resolution of 304 px × 240 px. Its 255 k labels consist of bounding boxes for pedestrians and cars provided at a frequency between 1 and 4 Hz. In the fashion of previous work (Li et al., 2022; Gehrig & Scaramuzza, 2023; Perot et al., 2020), we do not consider small bounding boxes with any side length $\leq 10$ px or a diagonal $\leq 30$ px. The minimal distance between bounding boxes in the dataset is 250 ms. Therefore we construct the sequence samples every 125 ms to ensure temporal alignment of time steps and labels.

Since it is a widely used metric for object detection, we report the mean Average Precision (mAP) over 10 IoU thresholds with the starting value 0.5, an increment of 0.05, and an ending value of 0.95.

**Implementation**    We adopt standard training strategies based on the implementation of Cordone et al. (2022), utilizing an AdamW optimizer with a weight decay of $1 \times 10^{-4}$ and implementing a cosine decay for the learning rate. Additionally, we train with a batch size of $14$ and an in initial learning rate of $4 \times 10^{-5}$. For the event data input, we consider sequences of length $s_{train} = 5$, where each time interval consists of a discretized 125 ms time window which is further divided into $T = 5$ time steps. Just like Cordone et al. (2022), we utilize the Focal Loss (Lin et al., 2020) for the classification task, since it efficiently addresses the class imbalance issue, as well as a smooth L1 loss for the box regression. Inspired by the work of Gehrig & Scaramuzza (2023), we study the influence of data augmentation ,including horizontal flipping and zooming in/out in Section 4.2.2.

### 4.2 ABLATION STUDY

In this section, we explore the key factors that considerably influence our approach's performance. We investigate the effect of sequence length during training and assess the influence of event data augmentation on training our SNN. The experiments are conducted on the Gen1 Automotive Detection Dataset, and evaluated using the best-performing models after 110 epochs.

Table 1: Evaluation of different training sequence lengths $s_{train}$ on various test sequence lengths $s_{test}$ of the Gen1 Automotive Detection Dataset.

| $s_{train}$ | $s_{test}$ | $\Delta t$ in ms | mAP | $AP_{50}$ | $AP_{75}$ | $AP_{car}$ | $AP_{ped}$ |
|---|---|---|---|---|---|---|---|
| 1 | 1 | 125 | 23.38 | 47.23 | 20.97 | 35.39 | 11.36 |
| 1 | 5 | 125 | 23.63 | 47.74 | 21.02 | 35.42 | 11.83 |
| 1 | full | 125 | 23.04 | 45.89 | 20.68 | 35.17 | 10.92 |
| 5 | 1 | 125 | 25.26 | 50.98 | 21.88 | 36.09 | 14.43 |
| 5 | 5 | 125 | 25.31 | 50.99 | 21.89 | 36.05 | 14.57 |
| 5 | full | 125 | 25.30 | 51.49 | 21.91 | 35.87 | 14.72 |

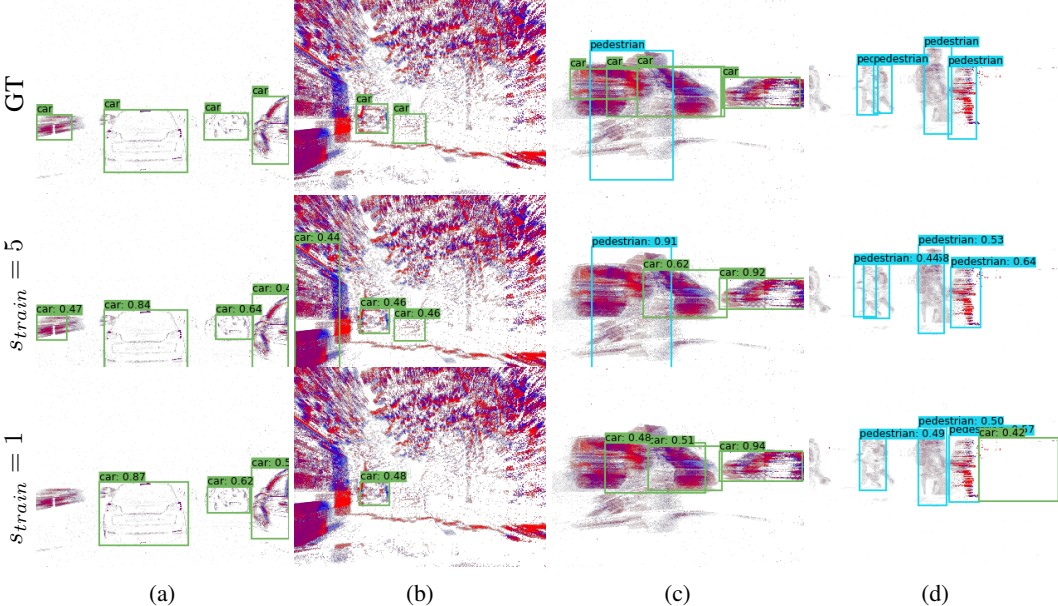

(a)    (b)    (c)    (d)

Figure 3: Visualization of the predictions on the Gen1 Automotive Detection Dataset with different training sequence lengths along with the ground truth bounding boxes, showing the network behavior in different scenarios (varying density of events and varying number of objects).

### 4.2.1 SEQUENCE LENGTH

Table 1 shows the results obtained by training our network on a single time interval $s_{train} = 1$ and a longer sequence length $s_{train} = 5$. We test the resulting networks on different sequence lengths $s_{test}$, including full sequences of the test split of the Gen1 Automotive Detection Dataset. A sequences is considered *full* if the maximum distance between labels is 200 time intervals. Sequences with higher distances are split in order to avoid excessive computation on data without any label. The data suggests a proportional performance increase corresponding to the length of the training sequence. This incline is observed across both object classes and different IoU thresholds. Training on longer sequences compels the network to prioritize the storage of important temporal information in its inner network state. The capacity of a single spiking neuron is limited, and during the processing of larger amounts of data, the network has to restrict the information flow in order to achieve the desired firing of the output neurons. A longer sequence can also contain multiple labels at different points in time, pushing the network to output correct predictions at any time. Therefore, even when tested on short sequences the network trained with $s_{train} = 5$ performs better. The data further shows that the performance of the $s_{train} = 1$ training decreases slightly when tested on the full sequence, indicating that the network is not able to fully leverage past temporal information because its membrane potentials are reset. Especially, its prediction for pedestrians dropped, posing a harder challenge due to more variety in movement. Figure 3 visualizes the results for the networks trained on sequence lengths 1 and 5 tested on *full* sequences. It can be seen that the network trained on

Table 2: Application of geometrical augmentation methods for the training on sequence length 5 and tested on *full* sequences.

| h-flip | zoom-in | zoom-out | mAP | $AP_{50}$ | $AP_{75}$ | $AP_{car}$ | $AP_{ped}$ |
|--------|---------|----------|-----|-----------|-----------|------------|------------|
|        |         |          | 25.30 | 51.49 | 21.91 | 35.87 | 14.72 |
| ✓      |         |          | 25.40 | 51.03 | 22.11 | 36.19 | 14.61 |
|        | ✓       |          | 26.85 | 53.41 | 23.63 | 37.80 | 15.90 |
|        |         | ✓        | 26.29 | 52.75 | 23.18 | 37.66 | 14.93 |
| ✓      | ✓       | ✓        | 26.88 | 52.79 | 24.65 | 37.90 | 15.86 |

Table 3: Comparison to State-of-the-Art on the test split of the Gen1 Automotive Detection Dataset. Information that was approximated or does not apply is marked with a star *. The results for the ⋄ marked papers were taken from Gehrig & Scaramuzza (2023). Furthermore, the object detection methods YOLOv3 and YOLOX have been presented by Redmon & Farhadi (2018) and Ge et al. (2021) respectively.

| Method | Backbone | Arch. | Det. Head | Seq | Aug | $\Delta t$ | mAP | Params (M) |
|--------|----------|-------|-----------|-----|-----|------------|-----|------------|
| Inception + SSD (Iacono et al., 2018)⋄ | CNN | CNN | SSD | ✗ | - | - | 30.1 | > 60* |
| RRC-Events (Chen, 2018)⋄ | CNN | CNN | YOLOv3 | ✗ | - | - | 30.7 | > 100* |
| YOLOv3 Events (Jiang et al., 2019)⋄ | CNN | CNN | YOLOv3 | ✗ | - | - | 31.2 | > 60* |
| RED (Perot et al., 2020) | CNN | RNN | SSD | ✓ | ✗ | 50 | 40.00 | 24.1 |
| ASTMNet (Li et al., 2022) | (T)CNN | RNN | SSD | ✓ | ✗ | * | 46.7 | > 100* |
| RVT-B (Gehrig & Scaramuzza, 2023) | Transformer | RNN | YOLOX | ✓ | ✓ | 50 | 47.2 | 18.5 |
| RVT-S (Gehrig & Scaramuzza, 2023) | Transformer | RNN | YOLOX | ✓ | ✓ | 50 | 46.5 | 9.9 |
| ODSNN (Cordone et al., 2022) | DenseNet121-24 | SNN | SSD | ✗ | ✗ | 100 | 18.9 | 8.2 |
| **S-SOD (ours)** | DenseNet121-24 | SNN | SSD | ✗ | ✗ | 125 | 23.38 | 8.2 |
| **S-SOD (ours)** | DenseNet121-24 | SNN | SSD | ✓ | ✗ | 125 | 25.30 | 8.2 |
| **S-SOD (ours)** | DenseNet121-24 | SNN | SSD | ✓ | ✓ | 125 | 26.88 | 8.2 |

longer sequences is more capable of detecting multiple entities even with a small number of events in one time interval, as in Figure 3a. Furthermore, it better differentiates between multiple object instances due to past temporal cues, as demonstrated in Figure 3d.

### 4.2.2 DATA AUGMENTATION

Following the example of Gehrig & Scaramuzza (2023), we examine the effect of the following randomized event data augmentations on the performance of our SNN: Horizontal flipping, zooming in, and zooming out. We randomly choose whether to use no augmentation, or flipping and/or zooming, and then apply this choice to a whole sequence. For any resizing operation we use exact nearest neighbor interpolation in order to keep the data in its binary format. To zoom in, we crop the sample randomly along its spatial dimensions and then resize it to the original resolution. For zooming out, we down-sample the whole sequence spatially and place it at a random location within a zero tensor of the original size. Our parameter choice with respect to the randomness follows the one of Gehrig & Scaramuzza (2023). Table 2 shows the results of the different augmentation methods applied to the training based on sequence length 5. We can see that augmentation plays an important role for the training of SNNs. Zooming is most effective, especially for pedestrians, which usually appear smaller than cars in real world data, possibly due to an increased focus on particular features of otherwise small objects.

### 4.3 BENCHMARK COMPARISON

In this section, we contrast our sequence-aware SNN approach **Sequence-SOD** with previous approaches which include some form of inner state representation, evaluating their performance on the Gen1 Automotive Detection Dataset. A summary of the outcomes is presented in Table 3.

The results already show, how Recurrent Neural Networks (RNNs) are able to vastly outperform traditional Convolutional Neural Networks (CNNs) in event-based object detection. This is due to their capacity to save temporal information in their inner state and using this information for future predictions. The results also show a gap between RNNs and SNNs in general, due to the challenges

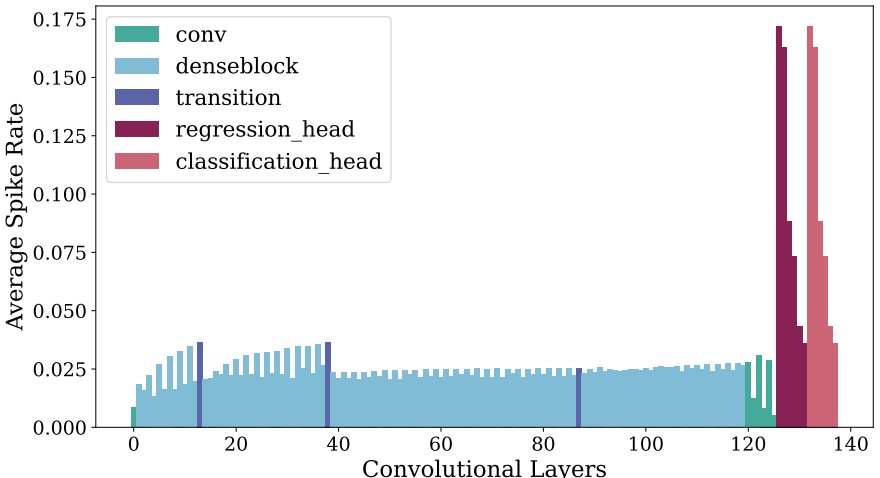

Figure 4: Average Spike Rates over all time steps for each convolutional layer.

of training the latter. In our work, we demonstrate for the first time the importance of a temporal inner state for SNNs. This can be seen by comparing our results to the only other spiking event object detector by Cordone et al. (2022). The numbers indicate that training on longer sequences indeed improves the performance by enforcing a meaningful inner state representation, which is able to discard dispensable information and prioritize significant temporal cues. Therefore, our method is able to set a new State-of-the-Art for SNNs in event-based object detection with an approximately 8% higher mAP.

## 4.4 ENERGY CONSUMPTION

Apart from their naturally suitable ability to process event data, a very significant advantage of SNNs is in terms of energy efficiency. In comparison to traditional ANNs, in which all neurons are activated for every iteration, SNNs have sparse activations. Furthermore, Multiply and Accumulate (MAC) operations for the synaptic weights are replaced by Accumulate (AC) operations in SNNs, since the spikes are represented only by zeros and ones.

Following the convention of previous work (Kim et al., 2022), we determine the energy consumption in terms of the total number of FLoating Point Operations (FLOPs) on standard CMOS technology. For a standard ANN, the FLOPs for multiplication and accumulation in a convolutional layer are both calculated by:

$$FLOPs_{conv,l}^{Mult} = FLOPs_{conv,l}^{Ac} = k^2 \times H_{out} \times W_{out} \times C_{in} \times C_{out}, \tag{3}$$

with $k$ as the kernel size, $(H_{out}, W_{out})$ as the output feature map height and width as well as $C_{in}, C_{out}$ as the number of input and output channels respectively.

Since spiking neurons only consume energy when a spike occurs, their FLOPs of a layer $l$ reduce based on the spike rate $R_l$. An average spike rate can be computed as the total number of spikes in a layer for all time steps divided by the number of neurons in $l$:

$$R_l = \frac{\sum_{t \in \{1,...,T\}, n \in N_l} S_t^n}{|N_l|}, \tag{4}$$

with $N_l$ as the set of neurons in layer $l$ and $S_t^n$ as the binary spike if neuron $n$ fired at time step $t$. The resulting spike rates per layer for our **Sequence-SOD** with $s_{train} = 5$ and $s_{test} = 1$ on the Gen1 Automotive Detection Dataset are shown in Figure 4. Averages over entire parts of the network are presented in Table 4.

Additionally, our network contains maxpool layers, which do not perform traditional floating-point operations. We estimate their FLOPs for ANNs and SNNs as an accumulation operation in the

Table 4: Energy consumption of an ANN and a SNN.

| Method | Network | $FLOPs^{Mult}$ | $FLOPs^{Ac}$ | $R_l$ | $E(mJ)$ | $E^{ANN}/E^{Method}$ |
|--------|---------|----------------|--------------|-------|---------|----------------------|
| ANN | Backbone | 2295.49 M | 2298.21 M | - | 10.56 | $1\times$ |
| | Heads | 351.32 M | 351.32 M | - | 1.61 | $1\times$ |
| | Overall | 2646.81 M | 2649.53 M | - | 12.17 | $1\times$ |
| SNN | Backbone | - | 2298.21 M | 0.0235 | 0.05 | $199\times$ |
| | Heads | - | 351.32 M | 0.0960 | 0.05 | $31\times$ |
| | Overall | - | 2649.53 M | 0.0296 | 0.10 | $\mathbf{116\times}$ |

following way:

$$FLOPs^{Ac}_{mp,l} = k^2 \times H_{out} \times W_{out} \times C_{in}. \tag{5}$$

We can now obtain the overall inference energy of an ANN ($E^{ANN}$) and of the equivalent SNN ($E^{SNN}$) across all layers $l$ of a network $N$:

$$E^{ANN} = \sum_{l \in N} FLOPs^{Mult}_{conv,l} \times E_{Mult} + \left( FLOPs^{Ac}_{conv,l} + FLOPs^{Ac}_{mp,l} \right) \times E_{Ac} \tag{6}$$

$$E^{SNN} = \sum_{l \in N} \left( R_l \times FLOPs^{Ac}_{conv,l} + FLOPs^{Ac}_{mp,l} \right) \times E_{Ac}, \tag{7}$$

where $E_{Mult} = 3.7$ pJ and $E_{Ac} = 0.9$ pJ are the energy consumed by a 45nm CMOS process for a 32bit Floating Point multiplication and accumulation operation respectively (Horowitz, 2014). The results of our analysis can be found in Table 4, showing a great superiority of the SNN compared to the equivalent ANN in terms of energy efficiency. It is important to note however, that the energy consumption of SNNs depends on the specific data set and the resulting spike rates in contrast to ANNs, which always execute the same number of operations. Moreover, SNNs can only achieve this level of energy efficiency, when deployed on neuromorphic hardware.

## 5 DISCUSSION AND LIMITATIONS

The training of SNNs on GPUs poses a great challenge, since GPUs are based on synchronous and parallel computing while the operations in SNNs are event-based. Therefore, the training can only be simulated using time steps and enforces a certain degree of accumulation of events into a more dense representation. This makes it especially difficult to train on long sequences of input data, because the gradient has to be computed over all time steps of a whole sequence, leading to a trade-off between training time and performance. Using longer sequences that contain multiple labels may result in a stronger training signal, however this signal will be distributed during backpropagation along the space and time dimensions of the network.

Furthermore, ANNs still suffer from lower accuracies compared to traditional ANNs and RNNs. Our method would greatly benefit from better training methods for SNNs, which are still an ongoing research topic. Still, SNNs can achieve a great reduction on energy requirements compared to equivalent ANN architectures.

## 6 CONCLUSION

We introduce a novel sequence-aware training approach for event-based object detection with SNNs. Our approach uses longer sequences, which include multiple labels at different points in time, during the training time in order to mimic real world usage. This allows the network to prioritize the storage of important temporal cues within its internal memory that is represented by the membrane potential of the nodes. This leads to an increased detection performance. Our network is further not only able to process short, single-label samples, but also longer sequences of continuous event streams with a high prediction frequency of 40 Hz. Future research avenues may explore the implementation of truncated backpropagation techniques in order to determine the full potential of very long training sequences while avoiding the great computational overhead of the full gradient.

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
