# OpenReview forum: "Sequence-SOD: Sequence-aware Spiking Object Detection for Event Cameras"
_ICLR.cc/2024/Conference — ICLR 2024 Conference Withdrawn Submission_

### Official Review · Reviewer_KHHd · 2023-10-14

**Soundness:** 2 fair
**Presentation:** 2 fair
**Contribution:** 1 poor
**Rating:** 3
**Confidence:** 3

**Summary:**

This paper proposes a Spiking Neural Network (SNN) based detector for event camera data called S-SOD. To leverage the temporal information of event sequences, S-SOD reuses the inner states of SNNs between time bins. Experimental results on Gen1 dataset show that S-SOD outperforms the SOTA SNN-based event detector, while still lagging behind RNN-based counterparts by a large margin.

**Strengths:**

- Temporal information is important for event-based object detection. The use of SNN's inner states is an intuitive way to leverage it
- Beating SOTA SNN-based detectors on Gen1

**Weaknesses:**

### Novelty
- The novelty of this work is limited. Both model architecture and training configs are the same as ODSNN. The only difference is the SNN module which reuses the inner states. Yet, this is just a minor change in my opinion. I would like to see more improvement, such as better SNN-based dense blocks instead of the naive one from ODSNN, or some SNN-based Transformer modules to better fuse temporal information

### Experiments
- The authors should conduct more experiments to validate their design. Only experimenting on one dataset is not enough. I would suggest the authors to test their method on 1MPx as well. Since the code of ODSNN has been released, it won't be very hard to compare with them on new datasets

**Questions:**

- The paper writing is unclear about how S-SOD reuses the inner states of SNNs. Does it reuse it across time bins (25ms) within a single time interval (125ms), or if it also reuse it across time intervals, i.e., keep reusing throughout the entire event sequence? Since objects in an event sequence might stop moving for more than 125ms, we need temporal information longer than a time interval to detect these objects. Therefore, I think only the latter case is reasonable
- It is still unclear to me how can S-SOD produce detections every 25ms instead of 125ms. The paper claims to "... by accumulating the network output over T time steps and dividing it by T", what does this mean? Are you running the SSD head to predict bboxes at the end of each time bin? Then what does accumlate+divide mean?

---

> ### Author Response · Authors · 2023-11-15
> **Response to Reviewer KHHd**
>
> Our sincere thanks to the reviewer for your thoughtful and important questions. We will answer them in the following:
>
> **Q1: Inner State reusage**
>
> We firstly want to clarify some terminology, since the usage of certain terms in the SNN literature is confusing. First of all, we talk about a 'sample' as the representation of (in our case) 125ms of events. Previous approaches only processed a single sample (e.g. of length 100ms), which was split into timesteps (which were sometimes further split into timebins like in  “Cordone et al. (2022)”).  These methods however process the timesteps of one sample with only one label and thus do not consider a longer sequence which include multiple samples with multiple labels at different points in time.
> As pointed out, it is only reasonable to consider long sequences of event data, meaning multiple 125ms samples. This is indeed the novelty we introduced in our training approach. Therefore, when we train on a sequence with length 5, our SNN processes 5 samples (= 25 timesteps) sequentially, which is equivalent to an overall time window of 625ms.
>
> We chose to use the same network design as “Cordone et al. (2022)”, since our technique can be applied to a multitude of network architectures and we wanted to ensure comparability. So far, no other SNN method for event object detection considered multi-label sequences during their training, showing that our approach is indeed novel. Our experiment further proved the importance of testing event object detectors in a real world setting, i.e. on continuous sequences of event data. Networks trained on single-label samples (mAP 23.38 (Table 1)) perform worse on sequences  (mAP 23.04 (Table 1)), making a sequence-aware training necessary.
>
> **Q2: Accumulation of timesteps**
>
> Thank you for pointing out the confusing description of our handling of timesteps. *We have improved the wording in the revised paper version (Section 3.2).*
> Our network indeed processes all the timesteps sequentially and therefore produces a prediction every 25ms. However, the average over multiple timesteps improves the overall prediction capability of the network. Therefore, the network indeed has to process the first 125ms before making the first prediction. Afterwards we can use a sliding window technique, which allows us to predict bounding boxes every 25ms for the rest of the sequence. The sequential processing of timesteps is one advantage of the SNN approaches over the current RNN methods (like "Recurrent Vision Transformers for Object Detection with Event Cameras (CVPR 2023)"), which feed all timesteps of one sample into the network at the same time.
>
> **Regarding additional experiments**
>
> Many SNN approaches often refrain from conducting experiments on large datasets such as the 1 Mpx Dataset due to their computationally intensive nature, particularly when training SNNs on GPUs. As a result, we have not yet conducted experiments on this dataset. With future advancements on the training of SNNS, e.g. on neuromorphic hardware, this gap will be closed.

---

> > ### Comment · Reviewer_KHHd · 2023-11-16
> > **Re: rebuttal**
> >
> > I thank the authors for the clarification.
> >
> > However, I still think the authors should conduct additional experiments on 1Mpx, and at least show some non-trivial results (and better results than the baseline ODSNN). I understand that 1Mpx is larger than Gen1 and has a higher labeling frequency, which may make the training slower. Still, I believe all event-based detection papers *published on top venues* have experimented on both datasets.

---

> > > ### Author Response · Authors · 2023-11-20
> > > **2. Response to Reviewer KHHd**
> > >
> > > While it's true that research in top-tier publications has looked into both the Gen1 and 1Mpx datasets, they've mostly focused on exploring RNNs and CNNs rather than SNNs. One key advantage of SNNs is their ability to handle smaller timesteps, enabling a higher prediction frequency. However, SNNs heavily rely on these timesteps to convey a broader range of outputs, given that neurons output only ones and zeros. Consequently, SNNs demand significantly more computational resources, especially on GPUs. This could explain why many leading SNN papers tend to avoid using extensive datasets like 1Mpx.

---

### Official Review · Reviewer_5brf · 2023-10-27

**Soundness:** 3 good
**Presentation:** 2 fair
**Contribution:** 2 fair
**Rating:** 5
**Confidence:** 5

**Summary:**

This paper proposes sequence-aware SNN for object detection, which processes long-term event streams and predicts bounding boxes wit a frequency of 40 Hz. Besides, the authors design an event augmentation strategy for SNNs in object detection task.

**Strengths:**

1) The relevant background knowledge of this paper is clearly explained.

2) The topic of sequence-aware spiking object detection for event cameras is very interesting topic.

3) This paper processes long sequences of the event stream data and predicts bounding boxes with a frequency of 40 Hz. In combination with a SSD network design, which are able to reach 26.88 mAP on the Gen1 Automotive Detection Dataset.

**Weaknesses:**

1) Innovation is only the representation of event data, and there is no innovation in the network structure, which does not fundamentally solve the problem of sequence-aware.

2) Compared with “Cordone et al. (2022)”, in addition to the modification of data representation, what is the innovation of the network structure?

3) The network structure is not explained clearly. Not explained clearly "maintaining the network's inner state within a sequence". For example, How is the red arrow in Figure 1 (b) implemented in the network?

**Questions:**

See weaknesses

---

> ### Author Response · Authors · 2023-11-15
> **Response to Reviewer 5brf**
>
> Thank you for your response and giving us the opportunity to remove all ambiguities. We decided to address all the mentioned weaknesses in a combined response, due to their relatedness.
>
> Indeed, our approach does not focus on the innovation of a new network design. For the sake of comparability, we decided to use the implementation by “Cordone et al. (2022)” , which, at the time of our submission, was the only other SNN approach for object detection on event data. Our sequence-aware training approach is generally independent of the network design and can be applied on top of most common architectures.
> Previous approaches like “Cordone et al. (2022)” process only single-label samples, which consists of 100ms of event data split into a certain number of timesteps. These timesteps are fed into the SNN sequentially, while the spiking neurons store their membrane potential in between timesteps. However, after each label the membrane potential of the neurons is reset to zero before the next event sample is processed. Our approach however trains on longer sequences of samples, meaning for example 5 * 125ms which are further split into timesteps of 25ms. Therefore,  "maintaining the network's inner state within a sequence" means that the membrane potentials of the spiking neurons are not reset after each label.
>
> *We also adapted the description in our paper in Section 3.1 and 3.2 to make these explanations more clear.*
>
>
> However, it needs to be pointed out that this is not a simple increase in the input size. The SNN still only processes the data of one timestep (25 ms)  at a time. Spiking neurons naturally posses the ability to be sequence-aware, since their membrane potential allows the storage and depiction of an inner state like RNNs. But as our experiments show, networks trained on single-label samples perform worse when tested on longer sequences with multiple labels at different points in time. Therefore, we believe that the training on sequences is necessary to enforce true sequence-awareness in SNNs. Future improvements in the network design may indeed facilitate this behaviour.

---

> > ### Comment · Reviewer_5brf · 2023-11-21
> > **I still stuck with the score.**
> >
> > I believe there are opportunities for enhancement in the original manuscript: 1) consolidating more detailed explanations of 1-2 innovations, preferably focusing on refining the network structure that integrates event characteristics, and 2) improving the clarity of the writing, specifically by providing a more detailed description of the network structure to aid the reader's understanding of the intricacies.In addition, I look forward to witnessing your work at a future top conference or in a top journal.

---

### Official Review · Reviewer_idF6 · 2023-10-30

**Soundness:** 3 good
**Presentation:** 3 good
**Contribution:** 3 good
**Rating:** 6
**Confidence:** 5

**Summary:**

This paper investigates the problem of object detection using Spiking Neural Networks (SNN) with event data as input. It analyzes the characteristics of membrane potentials of pulse neurons and uses continuous input data to train the SNN while maintaining the internal state of the SNN during this time to fully utilize the temporal information of events.

**Strengths:**

1. By combining the characteristics of event data and the advantages of accumulating membrane potential information in SNN, the article does not reset the membrane potential state of pulse neurons during training. It processes multiple consecutive samples and evenly divides the events within a fixed time interval in each sample as sequential input, replacing the time step of SNN, which improves the performance of SNN-based event data object detection methods.
2. The article validates the impact of data augmentation on the model and demonstrates the low power consumption of SNN through calculations.

**Weaknesses:**

1. The contribution 1 of the paper is not highly innovative as training SNN with continuous data is already common. For example, the method has been used in "Event-based Video Reconstruction via Potential-assisted Spiking Neural Network (CVPR 2022)," and when compared to the method in "Deep Directly-Trained Spiking Neural Networks for Object Detection (CVPR 2023)," the performance does not show a significant difference.
2. The paper claims the ability to process real-world data at 40Hz, but the paper's meaning is limited to events occurring every 25ms, which depends on the input settings. This may not be a significant contribution point, or the paper should test the overall inference speed in FPS to confirm real-time performance.
3. The data augmentation experiments and energy calculations are conventional and lack notable highlights. Additionally, the overall writing quality of the paper is not high, and it is recommended to review and correct numerous grammar errors.

**Questions:**

1. Clearly articulate how your method differs from existing ones, even if they share some similarities in terms of using continuous data; Highlight where your approach outperforms or provides unique advantages compared to existing methods.

2. To strengthen the claim of real-world data processing at 40Hz, you should clarify the conditions and constraints under which this performance is achieved. Specify the input settings and provide a broader context for this achievement.
Consider conducting experiments to test the overall inference speed in FPS across various real-world scenarios and datasets, not just under specific input settings. This will provide a more comprehensive evaluation of your model's real-time performance.

3. Review the paper for grammar errors.

---

> ### Author Response · Authors · 2023-11-15
> **Response to Reviewer idF6**
>
> Thank you for your questions and the mentioning of other related work, which will hopefully shed more light on the novelty of our approach. In the following we want to address the three questions posed:
>
> **Q1: Uniqueness of our approach**
>
> The amount of research on the topic of object detection for event data using SNNs is still very limited. Therefore, our approach is the first method taking into account the training on longer sequences regarding this specific application. Unfortunately, we were not able to include the paper "Deep Directly-Trained Spiking Neural Networks for Object Detection" as it was published at ICCV 2023, which took place after the submission deadline of the ICLR this year. Nonetheless, their approach focuses on a novel network architecture for event object detection, but does not include the training on sequences of events with multiple labels at different points in time. Like it is a common practice for SNN approaches, they process one sample (divided into timesteps) for each label separately. Since they apply an optimised network design compared to  "Object Detection with Spiking Neural Networks on Automotive Event Data (IJCNN 2022)", they are able to easily outperform them. We confidently assume, that combining their novel network architecture with our sequence-aware training would even further boost the object detection results.
> Regarding paper "Event-based Video Reconstruction via Potential-assisted Spiking Neural Network (CVPR 2022)",  the reconstruction of videos from event data depicts a different problem setting than object detection. The labels themselves are a sequence of images, which are captured at a fixed frame rate allowing for a training signal at fixed intervals. In object detection, however, there is no guarantee that an object and its label exist or are available at a certain point in time. Thus, our approach has to be able to handle the great variance in the timing of the training signal. Furthermore, our paper takes a more in-depth look into the affect of sequences. We showed the advantage of training on sequences with multiple labels at various points in time compared to single-label sample, especially on real world application setting, where full event streams have to be processed during test time (*23.04  mAP vs 25.30 mAP* (Table 1)). The work in "Event-based Video Reconstruction via Potential-assisted Spiking Neural Network (CVPR 2022)" did not focus on the advantages of sequence processing and thus did not conduct these evaluations.
>
> **Q2: Prediction Frequency**
>
> Indeed, the frequency of prediction is dependent on the input data and their representation. In the case of the, Gen1 Automotive Detection Dataset the time difference between labels is at least 125 ms, which we divided into 5 timesteps (= 25 ms). Our SNN processes one timestep at a time and then averages the output over the last 5 timesteps, which can be done using a sliding window implementation. Except for the first 125 ms, our approach is therefore able to produce a bounding box prediction for every  25 ms of input data, which in theory equals a frequency of 40 Hz. *We updated the information in Section 3.2 to express this more clearly.*
> Dividing the 125ms further, can indeed lead to an even higher prediction frequency. This does not consider the computation time on a GPU and the full speed advantage will only take an effect when implemented on neuromorphic hardware. However, we wanted to highlight the advantage of SNNs compared to the current RNN approaches (like "Recurrent Vision Transformers for Object Detection with Event Cameras (CVPR 2023)", since these process all timesteps at the same time, leading to a reduced prediction frequency.
>
>
> **Q3: Grammar**
>
> Thank you for pointing out the room for improvement regarding grammar. We revised the grammar and language throughout the whole paper.

---

> > ### Comment · Reviewer_idF6 · 2023-11-21
> >
> > I thank the authors' response.
> >
> > Overall, I believe this work integrates the temporal characteristics of SNN with the continuity of detection tasks, and the framework is innovative. However, I think the organization of the innovative aspects of the paper needs improvement. For example, Contribution 1 requires a more thorough analysis in relation to other tasks associated with SNN. Contribution 2 also needs a better explanation.
> >
> > I will maintain the current score.

---

### Official Review · Reviewer_N9Kv · 2023-10-31

**Soundness:** 2 fair
**Presentation:** 1 poor
**Contribution:** 1 poor
**Rating:** 3
**Confidence:** 4

**Summary:**

This paper proposes a full temporal SNN for spiking object detection, which achieves 26.88 mAP on the Gen1 Automotive Detection Dataset. However, their innovativeness is limited and the motivation for the study is not clear enough.

**Strengths:**

It seems to achieve good performance on the Gen1 Automotive Detection Dataset comparing the SNN-method.

**Weaknesses:**

1. This paper claims solved the problem of considering only a single, fixed-size sample of the event data in spiking object detection. However, the dataset Gen1 Automotive Detection Dataset used for experiments in the paper has multiple objects at the same time with variable sizes. The authors have not clearly stated the problem to be solved.
2. The main difference between spiking neurons and CNN neurons is the temporal memory through properties such as leakage, so the authors should explain how the "full temporal information" enhances SNNs.
3. Concerning the Sequence-SOD, it just increases the input event time domain length. The authors do not present any theory to support this approach and the method is too simple. As an academic paper, such contribution points are too few.
4. In conclusion, this manuscript needs to be further organized for motivation and to find the theoretical basis of the proposed method to inspire other scholars before it is published as a paper.

**Questions:**

Please See Weakness

**Details Of Ethics Concerns:**

Contribution points and their rationalization

---

> ### Author Response · Authors · 2023-11-15
> **Response to Reviewer N9Kv**
>
> We would like to thank you for providing your unique insight on the topic with us. We will address each of your concern separately:
>
>
> **Q1: Problem Statement**
>
> The Gen1 Automotive Detection Dataset is indeed very challenging, since it contains many objects of various sizes. Furthermore, the labels were generated based on the corresponding images captured by an RGB camera. Consequently, the bounding boxes are only available at certain times and with varying frequency. Previous approaches for object detection on event data using SNNs only use the events within a specific time window just before a labeled RGB frame. We used the term "single label" sample in this case, which means they only train on the labels in one RGB frame, which may include multiple bounding boxes. *In the revised version of our paper, we explained this situation now more clearly*. However, since the event data depicts a stream, the bounding boxes in consecutive RGB frames are temporally linked. This is especially crucial in event data, since non-moving objects do not produce events. Therefore, we train our network on longer sequences of event data which include multiple frames with multiple bounding boxes. Our experiments show that networks trained solely on 'single label' samples exhibit a drop in performance when applied to a continuous event stream, from 23.38 mAP to 23.04 mAP (Table 1). Our method is the first event object detector using SNNs which exploit longer sequences with multiple bounding boxes at various points in time during training, leading to an obvious *performance boost from an mAP of 23.04 to 25.30* (Table 1).
>
>
> **Q2: Difference to CNNs**
>
> As it was correctly pointed out, one great advantage of SNNs over CNNs is their membrane potential as well as its leakage over time. Spiking neurons accumulate the incoming spikes in their membrane potential over time. In our case, we unfortunately have to simulate the time in discrete timesteps, since we are training on a GPU. Nonetheless, we feed a subsequent event timesteps of size 25ms sequentially into the SNN, which has the advantage of a higher prediction frequency. The membrane potential can then be seen as an inner state representation of the network similar to Recurrent Neural Networks. Since conventional CNNs do not posses any memory they are not able to utilise the temporal correlation between consecutive inputs.
>
>
> **Q3: Input length**
>
> Indeed our network processes longer sequences of event data during training. However independent of the sequence length, we split the event data into 25 ms timesteps. These timesteps are processed sequentially by the SNN. A longer sequence of sequential input data forces the network to prioritise which information to store, since its storage capabilities are limited. This is different compared to a simple increase in the input data length, which would lead to a larger amount of data being fed into the network at the same time. Undesirable consequences of such an approach would be a low prediction frequency and therefore reduce the applicability in real world scenarios.
>
>
> **Q4: Inspiration to other scholars**
>
> Currently, the amount of spiking object detectors for event data is very limited, even though the energy efficiency of SNNs as well as Event cameras posses great potential for autonomous driving tasks. Our approach includes the first exploration of the sequence-aware training of SNNs for event object detection. We show, that networks trained on single-label samples have a decreased performance when applied to a full event stream, which would be the case in real world scenarios. Our hope is it, that our paper can inspire future research not only to incorporate sequence learning in their approaches, but also consider testing their performance for applications on full event streams.

---

### Comment · Area_Chair_VxK3 · 2023-11-15
**Please engage in reviewer-author discussion**

Dear reviewers,

The paper got diverging scores. The authors have provided their response to the comments. Could you look through the other reviews and engage into the discussion with authors? See if their response changes your assessment of the submission?

Thanks!
AC